# Shear stress-exposed pulmonary artery endothelial cells fail to upregulate HSP70 in chronic thromboembolic pulmonary hypertension

**William Salibe-Filho**[1]*, **Thaís L. S. Araujo**[2], **Everton G. Melo**[2], **Luiza B. C. T. Coimbra**[2], **Monica S. Lapa**[1], **Milena M. P. Acencio**[1], **Orival Freitas-Filho**[3], **Vera Luiza Capelozzi**[4], **Lisete Ribeiro Teixeira**[1], **Caio J. C. S. Fernandes**[1], **Fabio Biscegli Jatene**[3], **Francisco R. M. Laurindo**[5], **Mario Terra-Filho**[1]

1 Pulmonary Division, Heart Institute (InCor), Hospital das Clínicas da Faculdade de Medicina da Universidade de São Paulo - São Paulo, Brazil, 2 Department of Biochemistry, Institute of Chemistry, University of São Paulo, São Paulo, Brazil, 3 Cardiovascular Surgery Division, Hospital das Clínicas da Faculdade de Medicina da Universidade de São Paulo - São Paulo, Brazil, 4 Department of Pathology, Faculdade de Medicina da Universidade de São Paulo - São Paulo, Brazil, 5 Vascular Biology Laboratory, Heart Institute (InCor), Hospital das Clínicas da Faculdade de Medicina da Universidade de São Paulo - São Paulo, Brazil

* wsalibe@alumni.usp.br

**Data Availability Statement:** All relevant data are within the manuscript and its Supporting information files.

## Abstract

The pathophysiological mechanisms underlying chronic thromboembolic pulmonary hypertension (CTEPH) are still unclear. Endothelial cell (EC) remodeling is believed to contribute to this pulmonary disease triggered by thrombus and hemodynamic forces disbalance. Recently, we showed that HSP70 levels decrease by proatherogenic shear stress. Molecular chaperones play a major role in proteostasis in neurological, cancer and inflammatory/infectious diseases. To shed light on microvascular responses in CTEPH, we characterized the expression of molecular chaperones and annexin A2, a component of the fibrinolytic system. There is no animal model that reproduces microvascular changes in CTEPH, and this fact led us to isolated endothelial cells from patients with CTEPH undergoing pulmonary endarterectomy (PEA). We exposed CTEPH-EC and control human pulmonary endothelial cells (HPAEC) to high- (15 dynes/cm$^2$) or low- (5 dynes/cm$^2$) shear stress. After high-magnitude shear stress HPAEC upregulated heat shock protein 70kDa (HSP70) and the HSP ER paralogs 78 and 94kDa glucose-regulated protein (GRP78 and 94), whereas CTEPH-ECs failed to exhibit this response. At static conditions, both HSP70 and HSP90 families in CTEPH-EC are decreased. Importantly, immunohistochemistry analysis showed that HSP70 expression was downregulated in vivo, and annexin A2 was upregulated. Interestingly, wound healing and angiogenesis assays revealed that HSP70 inhibition with VER-155008 further impaired CTEPH-EC migratory responses. These results implicate HSP70 as a novel master regulator of endothelial dysfunction in type 4 PH. Overall, we first show that global failure of HSP upregulation is a hallmark of CTEPH pathogenesis and propose HSP70 as a potential biomarker of this condition.

**Funding:** This work was supported by Fundação de Amparo a Pesquisa do Estado de São Paulo (FAPESP) in the form of a research grant awarded to TLSA (18/13739-8) and fellowships awarded to TLSA (19/20435-8, 15/06210-2), LBCTC (20/11249-3) and EGM (19/25503-9), Centro de Pesquisa, Inovação e Difusão FAPESP (CEPID "Processos Redox em Biomedicina") in the form of a research grant awarded to FRML (13/07937-8), and Fundação Zerbini and Coordenação de Aperfeiçoamento de Pessoal de Nível Superior - Brasil (CAPES) (Finance Code 001) in the form of funds used for the purchase of materials and reagents. The funders had no role in study design, data collection and analysis, decision to publish, or preparation of the manuscript.

**Competing interests:** The authors have declared that no competing interests exist.

## Introduction

Chronic thromboembolic pulmonary hypertension (CTEPH) is a vascular disease, including group 4 of pulmonary hypertension (PH) characterized by an intraluminal thrombus that causes pulmonary artery obliteration resulting in right ventricular overload [1]. After at least three months of anticoagulation, the gold standard treatment for CTEPH is pulmonary endarterectomy (PEA) for eligible patients [2]. The second line treatment is medical therapy or balloon pulmonary angioplasty [1]. Despite hemodynamic improvement and increased median survival after PEA, it is remarkable that 35% of patients remain with pulmonary hypertension [3]. To date, the underlying microvascular biology of poor PEA responses as well CTEPH physiopathology is unclear, with such late-onset PH attributed to small-vessel arteriopathy [2, 4].

Hemodynamic forces in pulmonary circulation contribute to endothelial cell phenotype mainly through shear stress [5]. Overall physiological shear stress can vary from high (15 dynes/cm$^2$) to low (4 dynes/cm$^2$) in lung arterial tree [6, 7]. However, in CTEPH patients, shear stress may reach above 15 dynes/cm$^2$, especially in the unobstructed areas disrupting normal vascular homeostasis. The PH is tightly associated with CTEPH pathogenesis triggered by vascular remodeling [8]. The vessel tonus is mediated by endothelial cells in line with smooth muscle cell phenotype changes. Nitric oxide (NO) is most important vasodilator synthesized by endothelial nitric oxide synthases (eNOS), cytosolic endothelial protein, whose levels are decreased in PH led to increasing in pulmonary vascular resistance (PVR) [9].

eNOS activation depends on cytoplasmic 90kDa heat shock protein (HSP90). In basal condition, caveolin-1 maintains eNOS in the inactive form while HSP90 is associated with eNOS after endothelial cells stimulation with vascular epidermal growth factor (VEGF), estrogen, histamine, statins, and shear stress led to enhanced NO production [10]. Although NO depletion has been demonstrated in CTEPH patients [11]. eNOS, caveolin and HSP90 expression is unknown in any cell type lung from CTEPH patients.

Another important cytosolic HSP is 70kDa heat shock protein (HSP70) which cooperates with HSP90 in protein folding of some protein clients [12]. Intracellular HSP70 plays a protective role in cardiovascular diseases such as atherosclerosis [13], and endothelial cell Akt1 phosphorylation [14, 15], p110 PI3K subunits expression [15] and tube formation stimulated by VEGF have been demonstrated [15, 16]. HSP70 is known to be a mainstay in the endothelial cell response. Besides angiogenesis, HSP70 loss of function decreases VEGF-dependent EC migration and proliferation [15, 16]. Recently, we showed that HSP70 expression was decreased in human endothelial cells submitted to low laminar shear stress and in mice aortic arch which is exposed to pathological shear stress [17]. Therefore, HSP70 expression is modulated by hemodynamic patterns associated with the proatherogenic and atheroprotective flow [17]. Further, HSP70 has anti-inflammatory effects [18] and GRP78, an ER resident HSP70, exerts vascular protection through antithrombotic effects [19].

In order to better understanding CTEPH pathogenesis, we hypothesized that global failure in molecular chaperones could be present in CTEPH patients which in turn contributes to endothelial dysfunction observed in this severe pulmonary disease. Our study aimed to understand better the endothelial dysfunction involved in CTEPH patients, based on cytosolic and ER HSP70 and HSP90 expression. Furthermore, the endothelial cell response was analyzed through tube formation and migration. Finally, the pharmacological inhibition of HSP70 was evaluated. Overall, our data shed light HSP70 is a novel player in CTEPH disease and bring molecular chaperones as potential specific biomarker to this type 4 PH.

## Materials and methods

### Ethics statement

All participants enrolled in the present study signed informed consent before their inclusion.

This study was approved and followed the guidelines of the Ethics Committee for Analysis of Research Projects of the Clinics Hospital of the University of São Paulo School of Medicine (approval number 1.051.734).

### Participants

The thromboembolic material of 7 patients who underwent PEA at the Heart Institute (InCor-HCFMUSP, Brazil) was studied. The surgical specimens were immediately incubated with 0.9% saline and 1% penicillin. Patients using PH-specific medication or who had antiphospholipid syndrome were excluded to avoid external interference in endothelial function.

### Cell isolation and culture

Endothelial cells were isolated from thrombi obtained from patients who underwent PEA. These cells were referred to as the CTEPH-EC group. The tissues were cut into 2x2-cm sections and treated with 0.2% collagenase type II (Worthington) for 15 min at room temperature [20] followed by centrifugation for 5 min at 4˚C at 1500 rpm. This supernatant and the pellet suspended in a specific medium for endothelial cells, EGM-2MV (Clonetics, Lonza, cat. number CC-3202), were placed 25-cm$^2$ culture bottles (Techno Plastic Products AG- TPP, cat. number 90028) and incubated in 5% CO2 at 37˚C. The culture medium was changed 3 days after the procedure and then every 48 hours. Human Pulmonary Artery Endothelial Cells (Thermo Fisher Scientific, cat. number C0085C) were used as control cell cultures and referred to as the HPAEC group.

### Flow cytometry analysis

The cultures with endothelial morphology were marked with anti-CD31 (BD Pharmingen, Alexa Fluor 647, mouse anti-human CD31, cat. number 558094) and anti-CD90 (Abcam, PE, ab957000) [21]. Then, the cells were quantified using the flow cytometry method (BD Excalibur). For the apoptosis experiments, annexin V-FITC (Oncogene, cat # PF032) was used. The supernatant was centrifuged, and the marker was incubated. After, the cells were stained using the protocol described by manufacturing and were analyzed in the flow cytometry.

### Immunohistochemistry experiments

Histological sections were produced with 3-mm thickness containing representative samples of endothelium extracted from the thrombus of patients with CTEPH, which was previously stored at -80˚C. Transplanted lungs were used as control tissue by using pieces of the pulmonary artery of the donor. All the samples underwent immunohistochemical analysis for the identification of cytoplasmic HSP70 (Abcam, anti-Hsp70 antibody (3A3), #ab5439) and annexin 2 (Abcam, anti-annexin A2 antibody C-terminal, #ab185957), according to the manufacturer's protocol. Histomorphometric quantification of the expression of the HSP70 and annexin A2 markers and total area quantification randomly of 5 different fields was performed by using a microscope-coupled image analyzer. The system consists of an Olympus-5 camera coupled to an Olympus microscope, from which the images are viewed on a monitor and evaluated on a digital imaging system (Software Image Pro-Plus 6.0).

### Shear stress assay

The experiments were performed on a 100-mm plate [22]. The cells were maintained in an atmosphere at 5% CO2 at 37˚C. The shear stress system was maintained at 5 and 15 dynes/cm$^2$ for 24 hours to simulate the flows exerted in the pulmonary vessels [23]. One plate was sustained to static conditions and used as a control. After 24 hours, the following variables were analyzed: HSP70, HSP90, GRP94, alpha 5 integrin, annexin A2, and PDI by Western blot in the shear stress (5 and 15 dynes/cm$^2$) and static conditions. All samples were run on the same gel, and the best representative images were chosen to the figures.

The cells were lysed in 20 mM HEPES pH = 7.2, 150 mM NaCl, 1 mM EGTA, 1.5 mM MgCl2, 10% glycerol, and 1% Triton-containing proteases and phosphatase inhibitors for 30 min at 4˚C. Sonicated lysates were centrifuged for 10 min at 12,000 rpm at 4˚C, and the supernatant was frozen at -80˚C. Forty-sixty micrograms of total protein underwent SDS-PAGE, transferred onto a nitrocellulose membrane, milk blocked, and incubated with the indicated antibodies. The antibody dilution varied from 1:1000–1:5000, including anti-PDI (MA3-019, Thermo Scientific), anti-GRP78 (ab21685, Abcam), GRP94 (ab52031, Abcam), HSP70 (ab5439, Abcam), anti-HSP90 (sc-13119, Santa Cruz Biotechnology), anti-integrin alpha 5 (ab150361, Abcam), anti-caveolin-1 (Cell Signaling Technology), anti-annexin A2 (ab185957, Abcam), and anti-β actin (AC-74, Sigma).

### Wound healing assay

The wound healing migration assay was performed following previously published methods [24]. HPAEC and CTEPH-EC were seeded at 1.5x10$^5$ cells in 12-well-plate. For each cell type, we used one basal and other with 30μM VER-155008 and grow cells for 4h. Then, cells were washed twice with HEPES and serum-starved for 2 hours in EBM-2 medium (Lonza) with 30 μM VER-155008 or DMSO. We made one scratch lengthwise per well with a sterile P200 tip and cultivate cells in presence of EGM-2 medium (Lonza) with 30 uM VER-155008 or DMSO. Cells were incubated for 18 hours at 37˚C and 5% CO2. Images were acquired at time 0 and 18h with a Zeiss microscope and 10X objective. The wound area was determined with Adobe Photoshop CS6, and the migration ratio was calculated by the equation %Healed = [(Area of original wound—Area of wound during healing)/Area of original wound]X100 [25].

### Tube formation assay

Tube formation assays were performed based on experiments described previously [26]. Each well of a pre-cooled 48-well plate was briefly coated with 150 μL Matrigel matrix (Corning, Tewksbury, MA, USA) and allowed to polymerize for 5 minutes at room temperature followed by 30 minutes at 37˚C. Primary HPAECs or CTEPH-EC isolated from CTEPH patients (3 x 10$^4$ cells per well) were seeded to the coated plate and incubated for 5h30min in EGM-2 alone or EGM-2 containing 50μM VER-155008 in a humidified incubator at 37˚C with 5% of CO2. Five images were obtained from each well using TissueFAXS imaging system (TissueGnostics, Vienna, Austria).

### Statistical analysis

Data are reported as the mean ± standard error of the mean. One-way analysis of variance (ANOVA) with the Newman-Keuls post-hoc test was performed for comparisons between 3 or more groups; the unpaired Student $t$ test was used for comparisons between 2 groups. The analysis (#) was performed to compare the condition of static and low shear stress using

unpaired Student t test. In both cases, the level of significance was 0.05. All statistical tests were performed using GraphPad Prism 5.0 software (GraphPad Software, Inc., La Jolla, CA).

## Results

CTEPH-ECs were isolated from the pulmonary arteries of patients who underwent PEA (Fig 1A). CTEPH- EC presented as a monolayer composed of rounded cells whose morphology was altered, with a larger size than HPAEC under static condition (Fig 1B). After exposure to high laminar shear stress, their round phenotype was shifted to an elongated one (Fig 1B, 15 dynes/cm$^2$). Low laminar shear stress did not alter the phenotype of CTEPH-EC cells compared with static condition (Fig 1B, 5 dynes/cm$^2$). Flow cytometry analysis revealed the presence of CD31 and absence of CD90, endothelial and mesenchymal cells markers, respectively (Fig 1C). Of note, cell viability evaluated through apoptosis was not altered in the CTEPH-EC group compared with control HPAEC (Fig 1D).

CTEPH-EC had diminished expression of constitutively expressed cytoplasmic HSC70 (HSPA8) and stress-inducible HSP70 (HSPA1A/B), both collectively refereed here as HSP70, under static condition (Fig 2A). The high laminar shear stress in HPAEC upregulated HSP70 expression, whereas in CTEPH-EC HSP70 amount was slightly decreased. The low laminar shear stress had a similar effect on both cells. In static conditions, HSP90 was also downregulated in CTEPH-EC compared with HPAEC, while fibronectin receptor alpha5 integrin was unchanged (Fig 2B).

Further characterization of molecular chaperone levels in CTEPH-EC showed that the ER HSP90 member (GRP94) was decreased in static condition and was not affected by applied shear stress (Fig 3A). ER HSP70 member (GRP78/Bip) was upregulated by high and low laminar shear stress in HPAEC, while only low shear stress increases GRP78 expression in CTEPH-EC (Fig 3B). However, not all ER chaperones were affected in CTEPH-EC, since protein disulfide isomerase (PDIA1), an important redox chaperone catalyst, had no change independent of condition (Fig 3B).

To analyze pathophysiological implications of the above results, we performed immunohistochemistry for HSP70 and annexin A2 in CTEPH pulmonary arteries in CTEPH patients

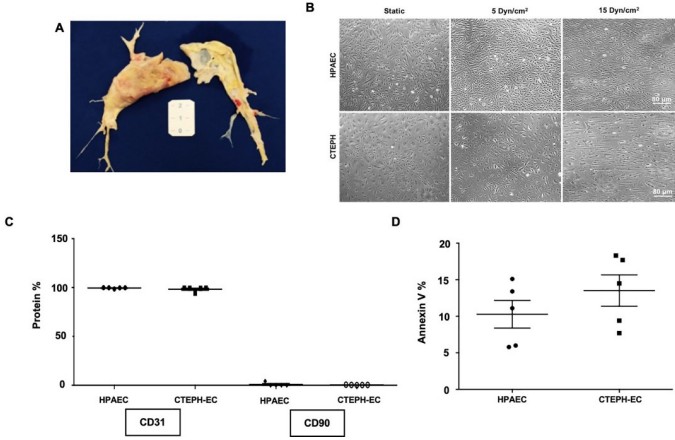

**Fig 1. Characterization of endothelial cells from CTEPH patient.** (A). Representative image from PEA samples. (B) Images of HPAEC and CTEPH-EC in static condition and after exposition to low (5 dynes/cm$^2$) and high shear stress (15 dynes/cm$^2$). (C). Flow cytometry analyses of CD31 and CD90 expression in HPAEC and CTEPH-EC (n = 5). (D) Annexin V quantification of HPAEC and CTEPH-EC in static condition (n = 5).

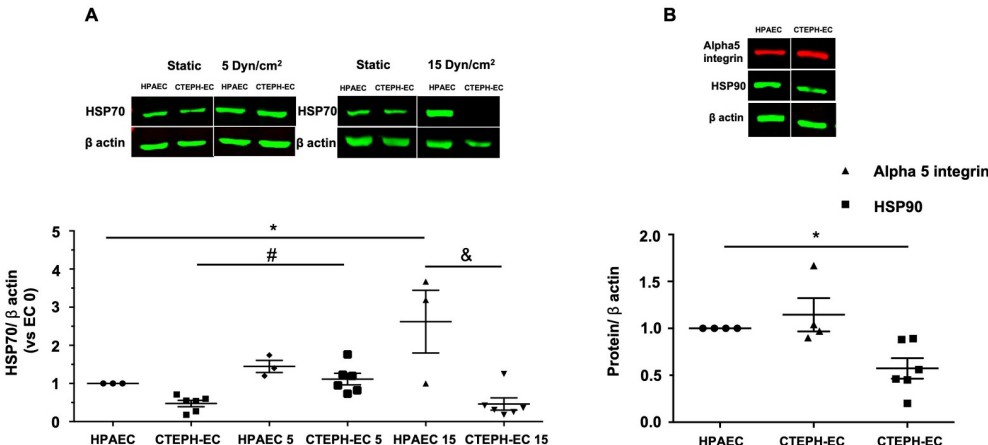

**Fig 2. Cytoplasmic protein expression in HPAEC and CTEPH-EC.** (A) Representative blot of HSP70 expression in static, low, and high magnitude shear stress (0, 5, and 15 dynes/cm$^2$). Graphs are presented as the mean ± standard error. (*) HPAEC vs HPAEC 15, (#) CTEPH-EC vs CTEPH-EC 5, (&) HPAEC 15 vs CTEPH-EC 15, $P < 0.05$. (n = 6–7). (B) HSP90 (n = 6) and alpha 5 integrin (n = 4) expression in static condition. Data are presented as the mean ± standard error. (*) HPAEC vs CTEPH-EC, $P < 0.05$. All of the samples were normalized to ß-actin and run in the same gel.

tissue using lung donors as a control. The results were close to those found in the CTEPH-EC, showing HSP70 downregulation [22% in controls vs 13% in CTEPH ($P < 0.05$)] (Fig 4A and 4B). In contrast, annexin A2 expression was upregulated in CTEPH [1.5% control vs 9.8% CTEPH (P < 0.05)] (Fig 4C and 4D).

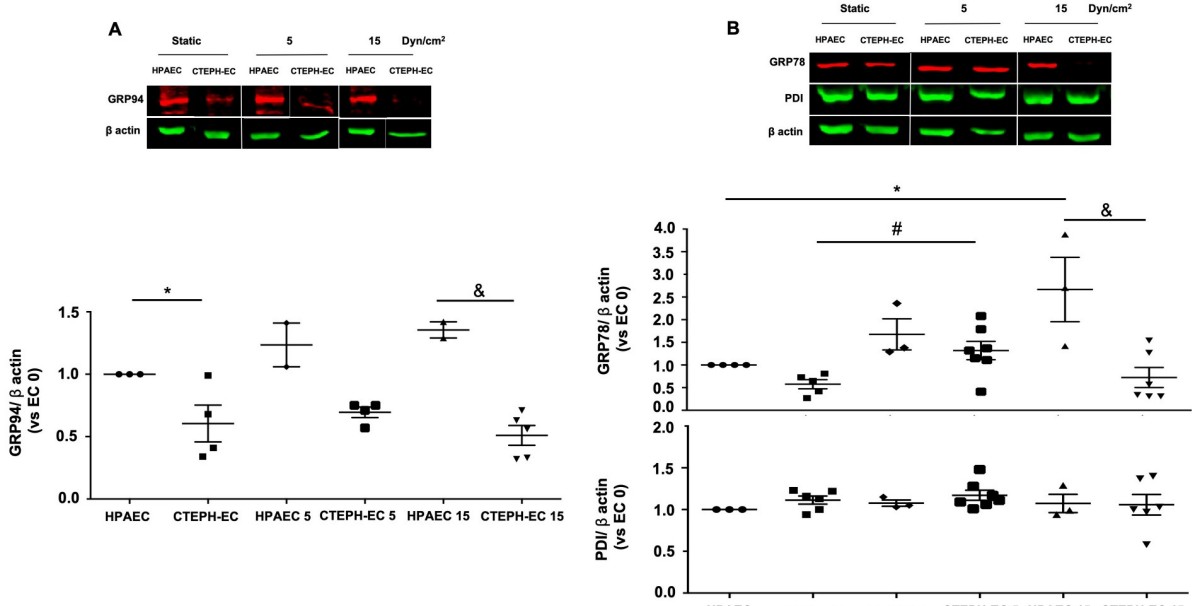

**Fig 3. Endoplasmic reticulum chaperones are downregulated in CTEPH-EC.** (A) Representative blot of the GRP94 expression in static, low, and high magnitude shear stress (0, 5, and 15 dynes/cm$^2$). Data are presented as the mean ± standard error: (*) HPAEC vs CTEPH-EC, (&) HPAEC 15 vs CTEPH-EC 15. (n = 5–7). (B) GRP78 and PDI expression before and after shear stress exposure as described in methods. Data are presented as the mean ± standard error: (*) HPAEC vs HPAEC 15, (#) CTEPH-EC vs CTEPH-EC 5 (&) HPAEC 15 vs CTEPH-EC 15, $P < 0.05$. (n = 6–7). All of the samples were normalized to ß-actin and run in the same gel.

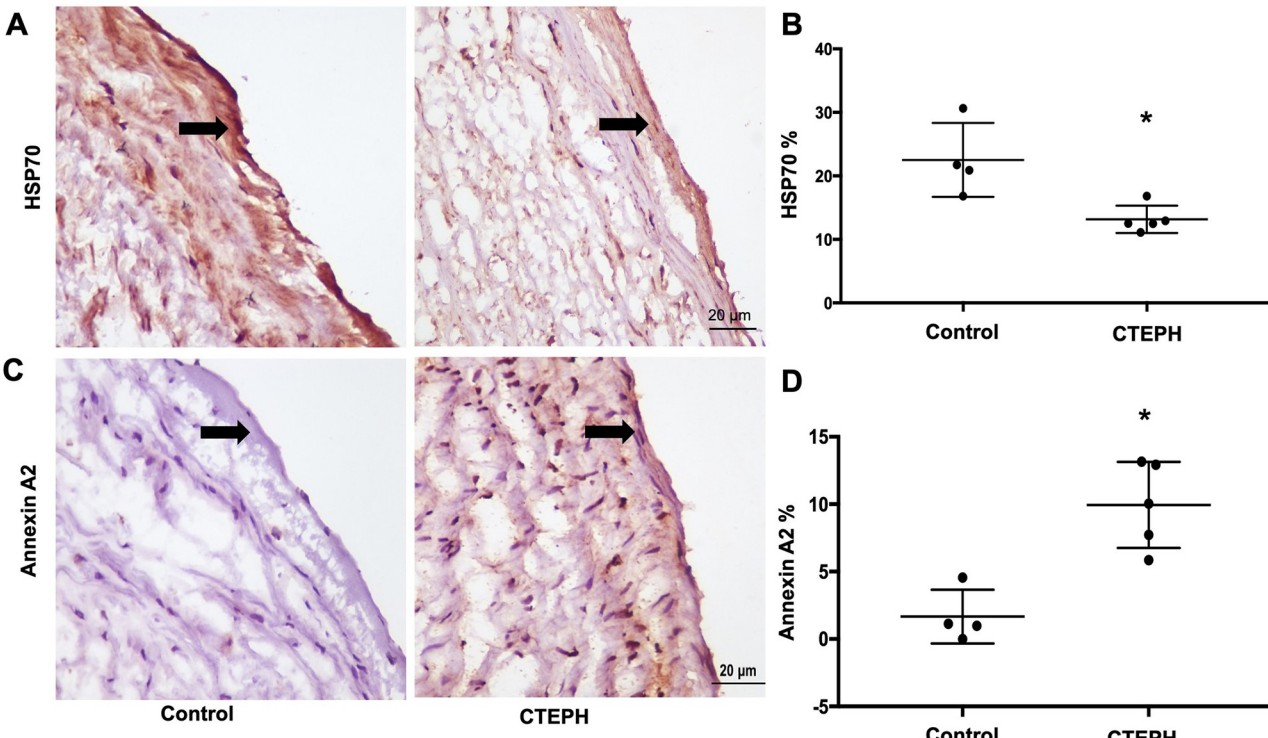

**Fig 4. Expression of HSP70 and annexin A2 in the internal part of the pulmonary artery from CTEPH patients and the pulmonary artery from lung transplant donors (control).** (A) Immunohistochemistry analyses of HSP70 and (C) Annexin A2 expression. (B, D) Quantification of images from (A) and (C), respectively. Data are presented as the mean ± standard error. Control: n = 4 // Patients: n = 5. (*) Control vs CTEPH, $P < 0.05$.

Endothelial dysfunction was proposed to be an essential component in CTEPH [4], however it has not been directly investigated in CTEPH patients. We took advantage of our successful isolation of endothelial cells to evaluate its physiological responses. Interestingly, CTEPH-EC presented strong impairment in migration (Fig 5) and tube formation (Fig 6), with less total branches and more cells in each node consistent with frustrated angiogenic response. Moreover, both responses were significantly decreased by the specific [27] HSP70 inhibitor VER155008 (Figs 5 and 6), clearly showing that HSP70 plays an essential function in migration and angiogenesis both in HPAEC and CTEPH-EC.

## Discussion

Our results shed a new light of molecular chaperones functions in CTEPH, showing novel insights into the pathophysiology of this disease condition. Overall, the collective downregulation of HSP70, HSP90, GRP94, and GRP78 expression in CTEPH-EC is a novel hallmark of endothelial cell dysfunction. Furthermore, in vivo HSP70 expression was decreased in CTEPH tissues, and endothelial function was shown to be supported by this chaperone. To the best of our knowledge, this report is the first to implicate HSPs, particularly HSP70, in CTEPH pathophysiology.

It has been suggested that CTEPH-related endothelial cells exhibit autophagy failure [21]. Together with our data, this further points to a possible disruption of proteostasis in this condition. We investigated HSP70, a molecular chaperone that works as an insular component in protein folding and also cooperates with HSP90, mediates protein folding or client stability [12]. Of note, both chaperones are the hub of a network sustaining cancer cell survival, in which

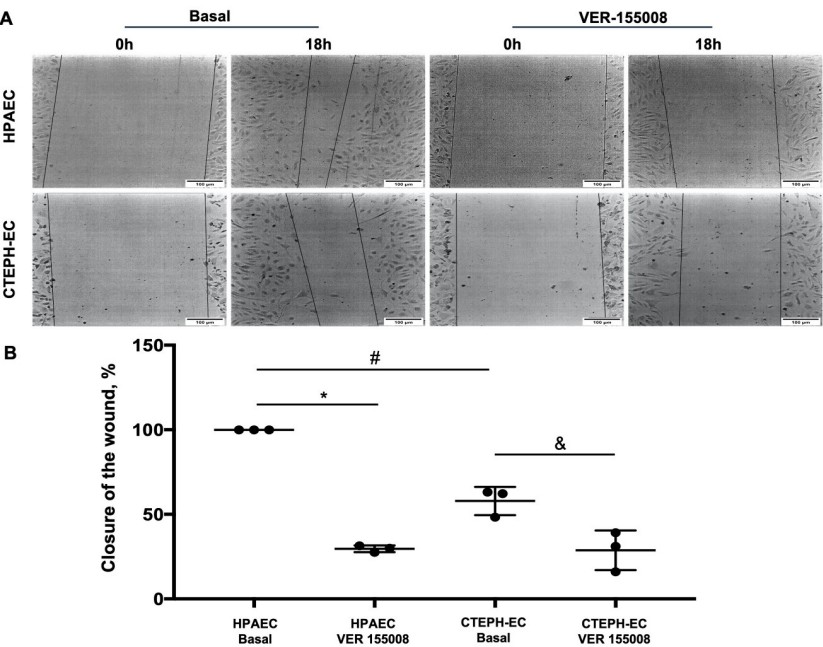

**Fig 5. Migration capacity of HPAEC and CTEPH-EC in basal and with HSP70 inhibitor VER-155008.** (A)
Representative images of HPAECs and CTEPH-ECs basal and after VER-155008. (B) Quantification of the migration.
Data are presented as the mean ± standard error: (*) HPAEC Basal vs HPAEC VER, (#) HPAEC Basal vs CTEPH-EC
Basal, (&) CTEPH-EC Basal vs CTEPH-EC VER, P < 0.05. (n = 3).

HSP90 inhibition has been explored as a basis for personalized treatment [28]. The possibility
of a protective role for the intracellular HSP70 family in cardiovascular diseases [29] opens the
possibility that therapeutic increases in the expression of this chaperone could mitigate endothe-
lial dysfunction. In CTEPH patient endothelial cells, the expression of HSP70 was substantially
downregulated in cell response to high shear stress (Fig 2A) and in vivo tissue expression (Fig
4), corroborating the findings of endothelial dysfunction and loss of protective effects.

ER stress is known to associate with monocrotaline-induced PH pathology [30]. Feaver
*et al*. have studied atherosclerotic lesions and ER stress, demonstrating that the shear stress to

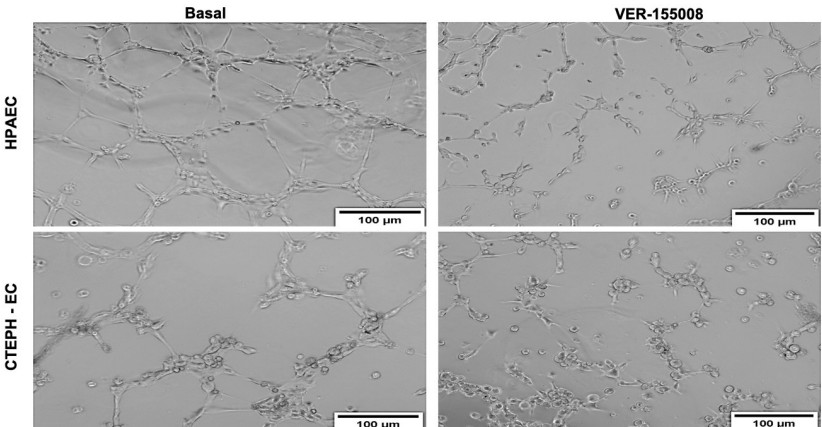

**Fig 6. Tube formation: Representative light microscopic images of the structures formed by HPAEC and
CTEPH-EC seeded and incubated for 5h30 min in the absence or presence of 50 uM VER-155008 (n = 6).**

which cells are exposed causes changes in the expression of GRP94 and GRP78 [19]. Here we found that the expression of GRP94 and GRP78 was decreased. These cells are unable to positively regulate these molecular chaperones in CTEPH-EC (Fig 3). These results suggest a possible specific CTEPH response and the decreased levels of HSP70 and HSP90 could be a possible trigger for disease development. These findings also provide a potential therapeutic option considering the availability of drugs capable of modulating the release of HSP70 and GRP78 [31, 32]. In Alzheimer's disease, a reduction of HSP70 has been demonstrated, and it has been proposed that increases in HSP70 levels induced by the YC-1 have protective effects [33]. Based on our results, upregulating HSP70 as a way to interfere with the vascular remodeling in CTEPH patients deserves further investigation.

Endothelial cell migration is a vital component of normal vascular development. Changes in this property are evaluated as an inadequate physiological response in pathophysiological situations [24]. Importantly, CTEPH-EC fails to upregulate HSP70 by high shear stress (15dynes/cm$^2$) which is deleterious in pulmonary circulation (Fig 2A). Furthermore, endothelial cell response analyzed though tube formation and migration (Figs 5 and 6) were strongly impaired in CTEPH-EC. Finally, CTEPH-EC dysfunction was enhanced by pharmacological HSP70 inhibition.

Others have shown that low shear stress is beneficial in pulmonary arteries [8]. Our data corroborate this view. Moser *et al.* [34] described the involvement of pulmonary circulation that goes beyond the mechanical obstruction. This phenomenon was later identified as a secondary pulmonary arteriopathy or small-vessel disease that was more evident in the unobstructed pulmonary arteries [4]. A study using an animal shunt model with hyperdynamic flow revealed that endothelial cell involvement led to disordered proliferation of pulmonary arterioles [35]. Our data open the possibility that changes caused by high flow in unobstructed areas also compromise the expression of HSPs, and failures of this positive regulation suggests that this arteriopathy promotes protein imbalance.

The changes in annexin A2 observed in our results deserve further discussion. Plasminogen activation to release plasmin on the cell surface is mediated by annexin A2 through direct binding of tissue plasminogen activator (t-PA) and plasminogen, which is synthesized by endothelium [36, 37]. Thus, annexin A2 translocation to the endothelial membrane is a key regulatory step in vascular fibrinolysis [38]. Antibodies against annexin A2 in antiphospholipid syndrome, together with inhibition of plasmin release, promote thrombus formation [39]; of note, 20% of CTEPH patients carry antiphospholipid antibodies [4]. Fibrinolysis defects are described in CTEPH pathophysiology, however, such changes remain controversial [40]. Upregulated annexin A2 expression (Fig 4) in our cells may represent an attempt to enhance fibrinolysis.

Together, our data showed for the first time that high-flow shear stress decreases the expression of heat shock proteins in endothelial cells of CTEPH patients. These results show that HSP70 failure supports endothelial dysfunction in these cells, suggesting that this chaperone may orchestrate a new pathophysiological pathway in CTEPH disease and highlight HSP70 as a new potential biomarker. These findings compose a new model to understand endothelial dysfunction in this disease. Microvascular involvement with a decrease expression in essential chaperones, such as HSP70 and HSP90, has been poorly explores and opens a new perspective in the understanding of this severe pulmonary vascular disease.

## Supporting information

**S1 File.**
(PDF)

## Acknowledgments

We thank Marcos Naoyuki Samano, MD, PhD, who provided the pulmonary arteries of lung donors.

## Author Contributions

**Conceptualization:** William Salibe-Filho, Thaís L. S. Araujo, Monica S. Lapa, Mario Terra-Filho.

**Data curation:** William Salibe-Filho, Thaís L. S. Araujo.

**Formal analysis:** William Salibe-Filho, Thaís L. S. Araujo.

**Funding acquisition:** Mario Terra-Filho.

**Investigation:** William Salibe-Filho, Thaís L. S. Araujo.

**Methodology:** William Salibe-Filho, Monica S. Lapa.

**Project administration:** William Salibe-Filho.

**Resources:** William Salibe-Filho, Thaís L. S. Araujo, Everton G. Melo, Luiza B. C. T. Coimbra, Milena M. P. Acencio, Orival Freitas-Filho, Vera Luiza Capelozzi, Lisete Ribeiro Teixeira, Caio J. C. S. Fernandes, Fabio Biscegli Jatene, Francisco R. M. Laurindo, Mario Terra-Filho.

**Supervision:** Mario Terra-Filho.

**Validation:** Thaís L. S. Araujo.

**Writing – original draft:** William Salibe-Filho, Thaís L. S. Araujo.

**Writing – review & editing:** William Salibe-Filho, Thaís L. S. Araujo, Everton G. Melo, Luiza B. C. T. Coimbra, Monica S. Lapa, Milena M. P. Acencio, Orival Freitas-Filho, Vera Luiza Capelozzi, Lisete Ribeiro Teixeira, Caio J. C. S. Fernandes, Fabio Biscegli Jatene, Francisco R. M. Laurindo, Mario Terra-Filho.

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
