## [Decision Letter · Decision Letter 0]

13 Nov 2019

PONE-D-19-24096

Shear stress-exposed pulmonary artery endothelial cells fail to upregulate HSP70 in chronic thromboembolic pulmonary hypertension

PLOS ONE

Dear Dr Salibe-Filho,

Thank you for submitting your manuscript to PLOS ONE. After careful consideration, we feel that it has merit but does not fully meet PLOS ONE’s publication criteria as it currently stands. Therefore, we invite you to submit a revised version of the manuscript that addresses the points raised during the review process.

We would appreciate receiving your revised manuscript by Dec 28 2019 11:59PM. To enhance the reproducibility of your results, we recommend that if applicable you deposit your laboratory protocols in protocols.io, where a protocol can be assigned its own identifier (DOI) such that it can be cited independently in the future. For instructions see: http://journals.plos.org/plosone/s/submission-guidelines#loc-laboratory-protocols

We look forward to receiving your revised manuscript.

Kind regards,

Michael Bader

Academic Editor

PLOS ONE

Journal Requirements:

Reviewers' comments:

Reviewer's Responses to Questions

**Comments to the Author**

1. Is the manuscript technically sound, and do the data support the conclusions?

Reviewer #1: Partly

Reviewer #2: Partly

2. Has the statistical analysis been performed appropriately and rigorously? 

Reviewer #1: Yes

Reviewer #2: Yes

3. Have the authors made all data underlying the findings in their manuscript fully available?

Reviewer #1: Yes

Reviewer #2: Yes

4. Is the manuscript presented in an intelligible fashion and written in standard English?

Reviewer #1: Yes

Reviewer #2: Yes

5. Review Comments to the Author

Reviewer #1: The authors showed a paradoxical reduction in the level of Hsp70 protein in shear-stress exposed pulmonary artery endothelial cells from chronic thromboembolic pulmonary hypertension. While their findings are interesting and intriguing, however, I have some concerns:

1) The authors claimed that the decreased hsp70 level was due to shear-stress induced proteostatic dysregulation. Though hsp70 is a major player in the protein quality control system, other proteins including CHIP, an E3 ubiquitin ligase play crucial roles. The authors cannot justify decreased hsp70 to increased protein degradation. If the authors want use the word proteostasis, they will have to back it up with data that decreased hsp70 expression is due to transcriptional or post transcriptional mechanism, both of which can be due to enhanced hsp70 degradation by ubiquitin-proteasomal signaling pathways.

2) The authors assessed the levels of Hsp70 in HPAECs and HPAEC from patients with chronic thromboembolic PH. However they failed to mention the specific isoform of hsp70. So far, there are about 12 -15 hsp70 isoforms, of which most of them are localized to the cytosol.

3) Importantly, what is the relevance of decreased hsp70 in the context of the pathogenesis of chronic thromboembolic PH?

4) The manuscript was written poorly with no flow to the story. It needs to be re-written and thoroughly edited for clarity.

Reviewer #2: this study from Saline-Filho et al evaluates the effects of shear stress on EC isolated from CTEPH patients compared to normal control PAEC. The role of proteostasis in pulmonary disease is in its infancy so studies which will increase our understanding of this are important for the field. However, this work has several major limitations that limit is potential impact on the field.

1. It is very descriptive and there is no mechanistic insight into why CTEPH PAEC cells down regulation molecular chaperones in response to flow. This needs to be addressed.

2. There are actually no studies presented that actually study proteostasis in the manuscript. It would seem that the authors need to evaluate at least a few targets of hsp70/hsp90 or other chaperones and see how these differ in EC isolated from CTEPH patients compared to normal control PAEC.

3. Can over or under expression of hsp70/hsp90 or other chaperones be used to restore proteostasis in EC from CTEPH patients or perturb it in normal control PAEC respectively? This is needed to demonstrate cause and effect relationships.

6. PLOS authors have the option to publish the peer review history of their article (what does this mean?). If published, this will include your full peer review and any attached files.

Reviewer #1: No

Reviewer #2: No

---

## [Author Response · Author response to Decision Letter 0]

14 Oct 2020

Dear Prof. Michael Bader,

 Please find attached to this electronic submission the revised version of our article entitled “Shear stress-exposed pulmonary artery endothelial cells fail to upregulate HSP70 in chronic thromboembolic pulmonary hypertension”, which I and my collaborators Thaís L. S. Araujo, Everton G. Melo, Luiza B. C. T. Coimbra, Monica S. Lapa, Milena M. P. Acencio, Orival Freitas Filho, Vera Luiza Capelozzi, Lisete Ribeiro Teixeira, Caio J. C. S. Fernandes, Fabio Biscegli Jatene, Francisco R. M. Laurindo, and Mario Terra-Filho are submitting for editorial consideration in PLOS ONE. 

 First of all, we would like to truly thank the reviewers and Editor for their time and effort to provide a thorough and constructive review of our work. We took great care to seriously consider and address all the concerns previously raised. The revised version now submitted has been quite extensively revised. The changes performed in our study have been described point-by-point in the "Answer to Reviewers" file and can be summarized by addition of two new experiments and extensive alterations in written manuscript based on suggestions of reviewers. Furthermore, the two new experiments were realized by Dr. Araujo and her students Everton and Luiza, the formerly dedicated efforts in performed angiogenesis assays while the latter dedicated to wound-healing assays. Now, we add their names in this revised version of our article. 

 The original blot images of the figures of the manuscript have been uploaded to the supporting information. 

 Overall, we believe these extensive modifications contributed to improve the paper and we hope it can now be suitable for publication in PLOS ONE. 

 We hereby expressly restate that this is fully original material not previously disclosed. 

 We are grateful for your attention to this article.

With best regards, 

 William Salibe-Filho

---

## [Decision Letter · Decision Letter 1]

13 Nov 2020

Shear stress-exposed pulmonary artery endothelial cells fail to upregulate HSP70 in chronic thromboembolic pulmonary hypertension

PONE-D-19-24096R1

Dear Dr. Salibe-Filho,

We’re pleased to inform you that your manuscript has been judged scientifically suitable for publication and will be formally accepted for publication once it meets all outstanding technical requirements.

Kind regards,

Michael Bader

Academic Editor

PLOS ONE

Additional Editor Comments (optional):

Reviewers' comments:

Reviewer's Responses to Questions

**Comments to the Author**

1. If the authors have adequately addressed your comments raised in a previous round of review and you feel that this manuscript is now acceptable for publication, you may indicate that here to bypass the “Comments to the Author” section, enter your conflict of interest statement in the “Confidential to Editor” section, and submit your "Accept" recommendation.

Reviewer #1: All comments have been addressed

Reviewer #2: All comments have been addressed

2. Is the manuscript technically sound, and do the data support the conclusions?

Reviewer #1: Yes

Reviewer #2: Yes

3. Has the statistical analysis been performed appropriately and rigorously? 

Reviewer #1: Yes

Reviewer #2: Yes

4. Have the authors made all data underlying the findings in their manuscript fully available?

Reviewer #1: Yes

Reviewer #2: Yes

5. Is the manuscript presented in an intelligible fashion and written in standard English?

Reviewer #1: Yes

Reviewer #2: Yes

6. Review Comments to the Author

Reviewer #1: Nice work! The authors should endeavor to determine the underlying mechanisms leading to decreased Hsp70 and Hsp90 expression in chronic thromboembolic induced-PAH. Identifying these mechanisms can provide therapeutic strategies in the treatment of PAH.

Also, the manuscript needs more work. The authors need today particular attention to tenses

Reviewer #2: Thea authors ahem adequately addressed the concerns raised in the prior review. No remaining concerns need to be addressed.

7. PLOS authors have the option to publish the peer review history of their article (what does this mean?). If published, this will include your full peer review and any attached files.

Reviewer #1: No

Reviewer #2: No

---

## [Editor Report · Acceptance letter]

24 Nov 2020

PONE-D-19-24096R1 

Shear stress-exposed pulmonary artery endothelial cells fail to upregulate HSP70 in chronic thromboembolic pulmonary hypertension 

Dear Dr. Salibe-Filho:

I'm pleased to inform you that your manuscript has been deemed suitable for publication in PLOS ONE. Congratulations! Your manuscript is now with our production department. 

Kind regards, 

on behalf of

Prof. Michael Bader 

Academic Editor

PLOS ONE